# Unsupervised vector-based classification of single-molecule charge transport data

Mario Lemmer[1], Michael S. Inkpen[1], Katja Kornysheva[2], Nicholas J. Long[1] & Tim Albrecht[1]

The stochastic nature of single-molecule charge transport measurements requires collection of large data sets to capture the full complexity of a molecular system. Data analysis is then guided by certain expectations, for example, a plateau feature in the tunnelling current distance trace, and the molecular conductance extracted from suitable histogram analysis. However, differences in molecular conformation or electrode contact geometry, the number of molecules in the junction or dynamic effects may lead to very different molecular signatures. Since their manifestation is *a priori* unknown, an unsupervised classification algorithm, making no prior assumptions regarding the data is clearly desirable. Here we present such an approach based on multivariate pattern analysis and apply it to simulated and experimental single-molecule charge transport data. We demonstrate how different event shapes are clearly separated using this algorithm and how statistics about different event classes can be extracted, when conventional methods of analysis fail.

[1] Department of Chemistry, Imperial College London, Imperial College Road, London SW7 2AZ, UK. [2] Institute for Cognitive Neuroscience, University College London, Alexandra House, 17-19 Queen Square, London WC1N 3AR, UK. Correspondence and requests for materials should be addressed to T.A. (email: t.albrecht@imperial.ac.uk).

The first single molecular conductance measurements were performed in the 1990s. Since then, a variety of different methodologies to measure charge transport across single molecules have been established, including fixed, chip-based electrode nanogaps, mechanical break-junctions and scanning probe microscopy techniques like scanning tunnelling (STM) and conducting tip atomic force microscopies[1–11].

For example, in STM-based tunnelling current–distance ($I(s)$) spectroscopy, as illustrated in Fig. 1a (refs 4,12,13), an STM tip at a constant bias voltage $V_{bias}$ is approached to the surface of a conductive substrate until a predefined set point tunnelling current $I_0$ is reached. The substrate typically carries the adsorbed molecule of interest, which is capable of forming a stable molecular bridge between the two electrodes via suitable anchor groups. The tip is then pulled away, while the tunnelling current $I$ is recorded as a function of tip/substrate distance $s$. For small $s$, $I$ is dominated by through-space tunnelling and decays exponentially with a characteristic decay constant $\beta$, Fig. 1b (region I)[14]. For larger $s$, charge transport through the bridge molecule is thought to dominate, implying that $I$ remains approximately constant at the plateau current $I_p$, namely until the molecular bridge is fully extended (region II). Further increase

in $s$ typically results in the rupture of the molecular bridge at some break-off distance $s_b$ and $I$ drops to the corresponding through-space value (effectively zero or noise level in most cases, region III). $I_p$ can then be related to the molecular conductance, while $s_b$ is a measure for the maximum length of the molecular bridge[15–17]. After collecting a sufficiently large number of $I(s)$ traces, histogram-based analysis is usually employed to extract the most probable values of $I_p$ and $s_b$ for a given molecular junction, that is from maxima in the histogram.

However, this approach is not without problem. First, as mentioned above, such an analysis relies on a particular signal shape (for example, a plateau feature) and hence an assumption about the expected outcome. If the signal shape is different or more complex, the meaning of a maximum in the corresponding histograms is less clear. Second, conventional histogram-based analysis has a strong focus on the most abundant class of signals (majority species) and sub-populations in the data may remain unnoticed.

To this end, it is now well-documented that even for seemingly simple molecular systems, such as Au/1,8-octanedithiol (ODT)/Au (Fig. 1a), the $I(s)$ data can contain a diversity of other shapes, such as slanted plateaus, non-linear and telegraphic noise

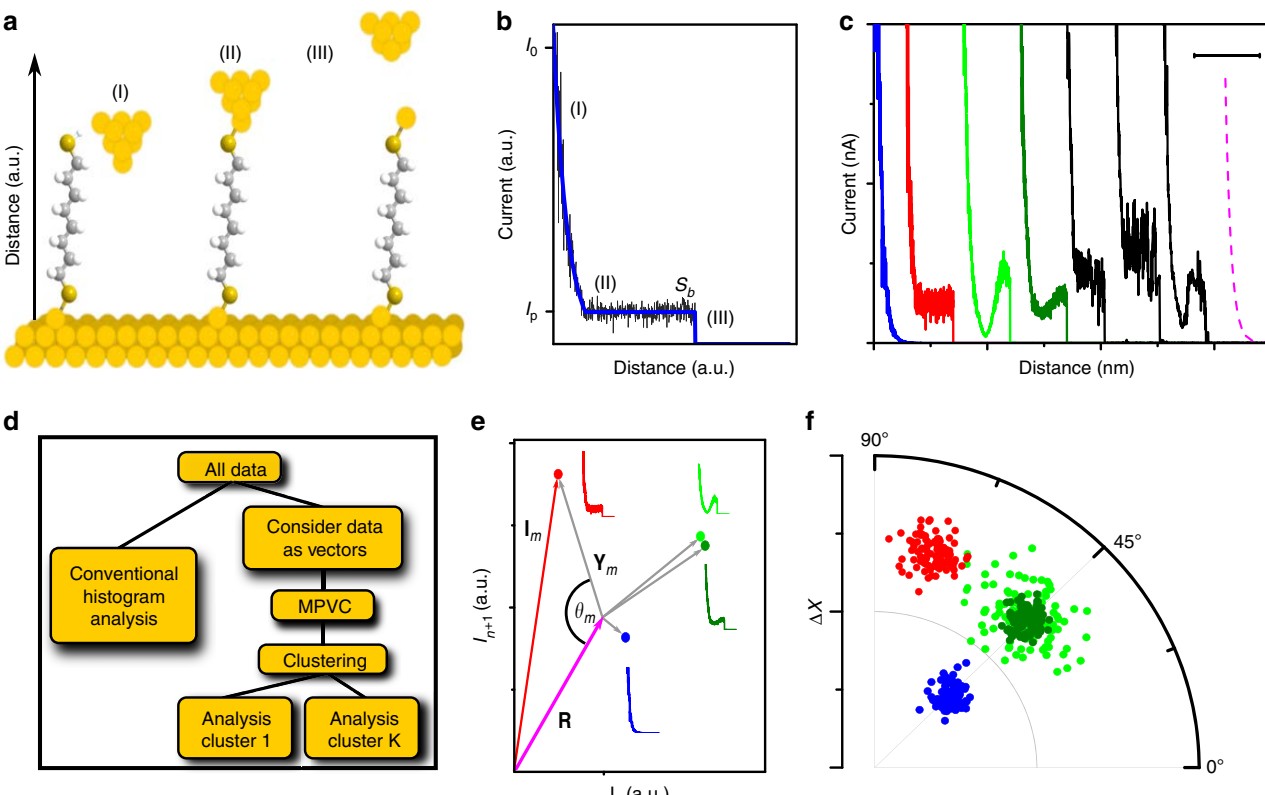

**Figure 1 | $I(s)$ spectroscopy and MPVC overview.** (**a**) Schematic of the $I(s)$ method with ODT in the junction, including three steps during junction formation and rupture. (**b**) Simulated $I(s)$ trace (blue) with added noise (black). Currents at (I) to (III) correspond to the three stages shown in **a**. (**c**) Blue, red, light green and dark green trace: Simulated $I(s)$ traces without plateau, with plateau and with non-linear plateau with high and low amplitude, respectively. Black traces: experimental traces for ODT. Dashed line ($\beta = 1.0\,\text{Å}^{-1}$): simulated reference vector **R**. Traces are offset for clarity. Scale bar, 1 nm. (**d**) Flow chart of the MPVC process, illustrating the philosophy behind MPVC. Conventional analysis is typically based on certain *a priori* assumptions regarding the shape of the $I(s)$ traces. This informs the methodology of the analysis, but can be misleading (see main text). MPVC: $I(s)$ data are regarded as vectors. Vector properties are calculated and used as classifiers for a measure of similarity between $I(s)$ traces; no prior assumptions are required. Similar traces are assigned to groups, based on suitable clustering algorithms. Physical interpretation of each group is the final step. (**e**) Vector representation of the four simulated traces shown in **c**, in two dimensions (using two data points from each trace) for illustration. In our data, each $I(s)$ trace has $N = 2,000$ data points, resulting in the same number of dimensions in vector space. (Simplified) distance vector **Y** and $\theta$ are shown for the plain plateau. The two green points represent two traces with similar $\Delta X$ and $\theta$ but different $h_r$. (**f**) Representation in cylindrical coordinates (view along $z$ axis), in principle involving all $N$ dimensions in the data. Red, blue and green clusters illustrate different conductance clusters, separable by $\Delta X$ and $\theta$. The light and dark green data points feature similar $\Delta X$ and $\theta$, but different $h_r$ values.

features, Fig. 1c (black traces)[11,12,18–23]. These may, for example, reflect the dynamics of the electrode surfaces, the molecular binding configuration, bond formation or rupture, molecular conformational changes, multiple (potentially transient) contact points between the electrodes and the molecule, and the presence of varying numbers of molecules in the junction[11,12,24–31]. Identifying and analysing these features of the I(s) trace is thus essential for a more complete fundamental understanding of the processes on the nanoscale.

Hence, there is a need to develop statistical tools for the analysis of single-molecule charge transport data that (i) do not make any a priori assumptions with regards to the signal shape and assess similarity within a given data set and (ii) allow for ideally unsupervised analysis and classification of large data sets, in order to capture the statistical complexity of the molecular system, including event classes that occur with low probability.

Here we present such a methodology, based on a new multi-parameter vector-based classification process (MPVC). We demonstrate its capabilities for a diverse set of simulated, but realistic I(s) data, as well as actual experimental results for two molecular systems, namely ODT and OPE (α,ω-dithiol terminated oligo phenylene(ethynylene)). Importantly, we show how MPVC is capable of identifying and extracting sub-populations in the data, where conventional methods of analysis fail.

## Results

**Vector-based data analysis.** Vector-based classification methods are powerful tools for categorizing the data and have found widespread application in such fields as genetics, robotics and neuroscience[32,33]. Generally, they operate by regarding a data set, for example, an I(s) curve with $N$ current values in the present case, as an $N$-dimensional vector $\mathbf{X}_n$ ($n = 1...N$); a total number of $M$ observations thus results in a data matrix $\mathbf{X}_{n,m}$ ($m = 1...M$) or, in short $\mathbf{X}_m$ (dropping $n$ for convenience). The Euclidean distance $|\Delta X|$ between the two vector points may then be used as a measure for similarity between different data sets $m$ and $m'$, equation (1):

$$\left|\Delta X_{m,m'}\right| = |X_{m'} - X_m| = \frac{1}{K \cdot \sqrt{N}} \cdot \sqrt{\sum_n \left(\mathbf{x}_{n,m'} - \mathbf{x}_{n,m}\right)^2} \quad (1)$$

where $K$ is an optional normalization constant (if required, so that $0 \leq \Delta X \leq 1$). If $\mathbf{X}_m$ and $\mathbf{X}_{m'}$ are identical, then $\Delta X_{m,m'}$ is zero; if they are very different, $\Delta X_{m,m'}$ is large. After calculating all combinations of distances between the $M$ data sets, a distance or probability criterion may then be used to classify the data. The computational effort scales with $(M - 1) \cdot M$, according to the variation formula, and can be very significant for some of our larger data sets (for example, involving 70,000 traces, see below).

In the case of I(s) data, however, we found the mutual distance criterion to be insufficient in many cases. A different, somewhat expanded methodology was required, which as we show below, significantly improved the classification performance.

In a first step, we defined an arbitrary, $N$-component reference vector $\mathbf{R}$, which generally depends on the data to be analysed. It could be determined self-consistently, based on some optimization parameter (for example, to maximize the variance of the existing data around $\mathbf{R}$). We chose a vector with noise-free, exponentially decaying current–distance values, which is similar to the experimental data without molecular binding events ($I_0 = 20$ nA; decay coefficient $\beta = 1$ Å$^{-1}$), Fig. 1c (dashed, magenta line).

We then calculate three vector properties in relation to the data sets $\mathbf{X}_m$ and $\mathbf{R}$, namely, the length of the difference

vector $\Delta X = |\mathbf{Y}_m| = |\mathbf{X}_m - \mathbf{R}|$; the angle $\theta_m$ between $\mathbf{Y}_m$ and $-\mathbf{R}$, equation (2),

$$\cos(\theta_m) = -\frac{\mathbf{R} \cdot \mathbf{Y}_m}{|\mathbf{R}| \cdot |\mathbf{Y}_m|} \quad (2)$$

and the reduced Hamming distance $h_\mathrm{r}$ between vectors $\mathbf{Y}_{\mathrm{r},m} = \mathbf{Y}_m/\mathrm{abs}(\mathbf{Y}_m)$ and $\mathbf{R}/\mathrm{abs}(\mathbf{R})$. The Hamming distance is the minimum relative number of component exchanges that render two vectors identical[34]. That is, if two vectors are identical, then the Hamming distance is 0 (no change required); if all components differ, then the Hamming distance is 1 (every component in one vector needs to be changed). Note that $\mathrm{abs}(\mathbf{Y}_m)$ and $\mathrm{abs}(\mathbf{R})$ are vectors containing the absolute component values of $\mathbf{Y}_m$ and $\mathbf{R}$, that is, they are not the length of the vectors $\mathbf{Y}_m$ and $\mathbf{R}$, respectively. The component values of $\mathbf{Y}_{\mathrm{r},m}$ and $\mathbf{R}/\mathrm{abs}(\mathbf{R})$ are thus either $-1$, 0 or 1.

In relation to the I(s) data, these three parameters $\Delta X_m$, $\theta_m$ and $h_{\mathrm{r},m}$ may be illustrated in the following way: $\Delta X_m$ is a measure for the total variation of a given I(s) trace relative to the reference trace $\mathbf{R}$, as described above. No distinction is made between positive and negative deviations from $\mathbf{R}$ and thus two differently shaped I(s) traces could feature the same $\Delta X_m$ value. For example, current values in a given trace 1 may be slightly higher than $\mathbf{R}$ over all $s$, while values in a second trace 2 may be significantly higher than $\mathbf{R}$ for low $s$ and then much lower than $\mathbf{R}$ at large $s$. However, the dot product between $\mathbf{R}$ and $\mathbf{Y}$ helps to differentiate these two cases, since positive and negative vector components cancel out. Thus, $\mathbf{R} \cdot \mathbf{Y}_1 \neq \mathbf{R} \cdot \mathbf{Y}_2$, resulting in different angles $\theta_1$ and $\theta_2$ between $\mathbf{Y}_1$ and $\mathbf{Y}_2$, and $\mathbf{R}$ in vector space (the denominator in equation (2) serves to normalize with regards to the vector lengths). Finally, $h_\mathrm{r}$ helps to quantify, how large a fraction of the I(s) values in a given trace lies above or below $\mathbf{R}$. For example, a third I(s) trace may oscillate around $\mathbf{R}$ in a way that $\Delta X_2 = \Delta X_3$ and $\theta_2 = \theta_3$, but with more data points above $\mathbf{R}$ than trace 2 (for example $\mathbf{Y}_{\mathrm{r},3}$ has more '1's than $\mathbf{Y}_{\mathrm{r},2}$). Thus, $h_{\mathrm{r},2} \neq h_{\mathrm{r},3}$ and the two curves can be differentiated.

It should be stressed that the above choice of parameters, while sensible, is neither complete nor exhaustive. Other statistical properties of the I(s) traces, such as the centre-of-mass of the curve, could be used to in addition or instead of one of the three parameter defined above. Generally, if the number of classifiers is too small, the classification will lack specificity and a cluster in the multi-parameter representation will contain many different types of I(s) traces. On the other hand, a too large number of classifiers may result in a large dispersion in the multi-parameter representation, rendering the identification of clusters more difficult. We found that the combination of $\Delta X_m$, $\theta_m$ and $h_{\mathrm{r},m}$ struck a good balance in this regard.

An important feature of the above MPVC algorithm, compared with conventional methods of analysis, is that it does not make any a priori assumptions with regards to the signal shape, for example, whether a plateau feature is present or not. Rather, it looks for similarities between the measured data sets, relative to $\mathbf{R}$ and based on the classifiers used. A large number of similar traces will thus produce regions of high point density in the $(\Delta X, \theta, h_\mathrm{r})$ representation, which can then be clustered and processed for further analysis.

Notably, MPVC does not provide the physical interpretation. It is partly for this reason that we chose both complex simulated data and well-characterized experimental system, where the interpretation is (largely) known, as a benchmark for MPVC here. The former allows for detailed characterization of the classification results (misclassification, especially where there are multiple groups), while the latter facilitates the physical interpretation of the clusters.

Finally, we note that by using a common reference vector we avoid having to compute the pairwise distance matrix for all $I(s)$ traces to determine similarity between traces. This effectively reduces the dimensionality of the data and is a significant advantage for large data sets (as for the ODT data with 70,000 traces, see below).

In the three-dimensional ($\Delta X$, $\theta_m$, $h_{r,m}$) representation, a range of different clustering algorithms can be applied to group the data, including $k$-means and $k$-medoids clustering, distribution- or density-based algorithms[35], Gaussian mixture models or neural networks[36,37]. A flow chart of the MPVC is provided in Fig. 1d. In the present work, we first use a distribution-based approach to illustrate the contribution of each parameter, and then employ three-dimensional Gustafson–Kessel fuzzy clustering in the second part of the paper[38].

Since both modulus $\Delta X$ and angle $\theta$ emerge from the classification process, we found representation in polar (or rather cylindrical) coordinates most convenient, Fig. 1f. The resulting event clusters can possess different and characteristic shapes, depending on the underlying nature of the variation in the data (say, a normally distributed plateau height or telegraphic noise features during bridge formation) and the reference vector. This in turn affects the way the individual clusters are identified and bound. For example, in cases where the cluster shape is symmetric, a Gaussian fit of the $\Delta X$, $\theta$ or $h_r$ histograms is usually most straightforward and event clusters may be defined based on a probability criterion (for example, within $\pm 2\,\sigma$ (95%) of the distribution mean). When the clusters are highly stretched or even semi-circular, which we found to be the case for the simulated data with normally distributed $s_b$, density-based clustering methods may be preferable[35].

To enhance the separation between event clusters, in particular at low (S/N) ratio, it can sometimes be advantageous to exclude segments of the $I(s)$ curve and define a region-of-interest (RoI), for example, where through-space tunnelling is dominant (at small $s$) or where the current is unlikely to contain well-defined, molecule-related information ($s \gg$ molecular length). This also reduces the computational cost, as we show below for a data set containing 70,000 $I(s)$ traces.

## Results

**Simulated data**. We first apply the MPVC algorithm to simulated data that resemble measured experimental data closely, in terms of current magnitude, decay coefficient and noise, and investigate the effect of current noise and break-off distance (variation) on the classification performance. In this way, we can also easily determine the misclassification rate, depending on the event and noise characteristics. We compare the results with conventional methods of analysis, namely one-dimensional (1D) current and two-dimensional (2D) current–distance histograms, which turn out to fail, if multiple event classes are present (as expected).

With regards to the effect of current noise, three data sets with different noise levels were generated, $cf.$ Methods section. Briefly, each data set consisted of 1,000 $I(s)$ traces, 80% of which were plain exponential decays (blue data points in Fig. 2) and 20% contained plateaus ($I_p = 1\,nA$, red data points; the reference vector $\mathbf{R}$ was calculated using $\mathbf{I}(s) = I_0 \cdot \exp(-\beta \cdot \mathbf{s})$ with $\beta = 1$ Å$^{-1}$, $I_0$ 20 nA (Note: the effect of the reference vector is demonstrated in Supplementary Fig. 1). A RoI was defined from 0.1 to 2 nm and $\Delta X$, $\theta$ and $h_r$ calculated as described above. For comparison, we show the 1D all-data current histograms in Fig. 2d–f.

At low noise (10% $\cdot I_p$ s.d., STDEV), both MPVC and current-histogram-based methods show excellent performance: the polar plot clearly shows two distinct event clusters that can easily be separated based on $\Delta X$ and/or $\theta$ (Fig. 2a). The all-data current histogram features a clear peak corresponding to $I_p$ and the extraction of the molecular conductance is straightforward. At an intermediate noise level (30% $\cdot I_p$), some separation is still achieved in the polar plot (Fig. 2b), based on $\Delta X$ and $\theta$. The all-data current histogram displays a peak on top of an exponentially decaying background; its position can be found by appropriate data fitting.

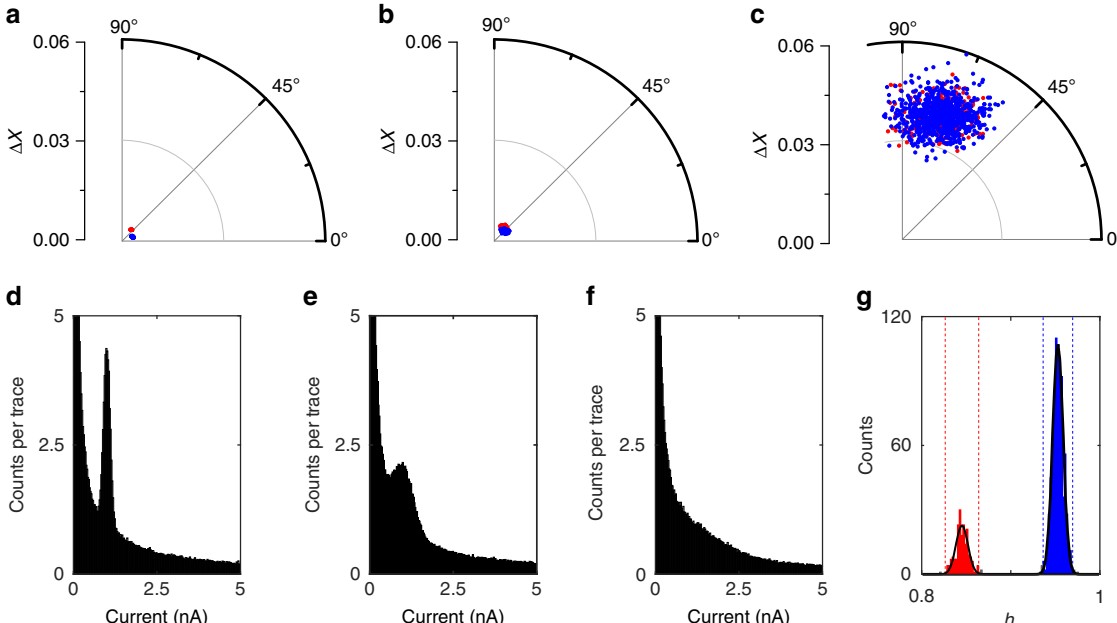

**Figure 2 | Simulated data with different noise levels.** (**a**–**c**): Polar plots of data sets with increasing noise levels (10%, 30% and 100% STDEV, relative to the $I_p$ (1 nA), respectively). Blue cluster: exponentially decaying, featureless $I(s)$ traces; red cluster: plateau-containing traces. (**d**–**f**): All the data 1D current histograms (bin width: 0.025 nA). (**g**) $h_r$-histogram of data set with 100% STDEV (bin width: 0.002).

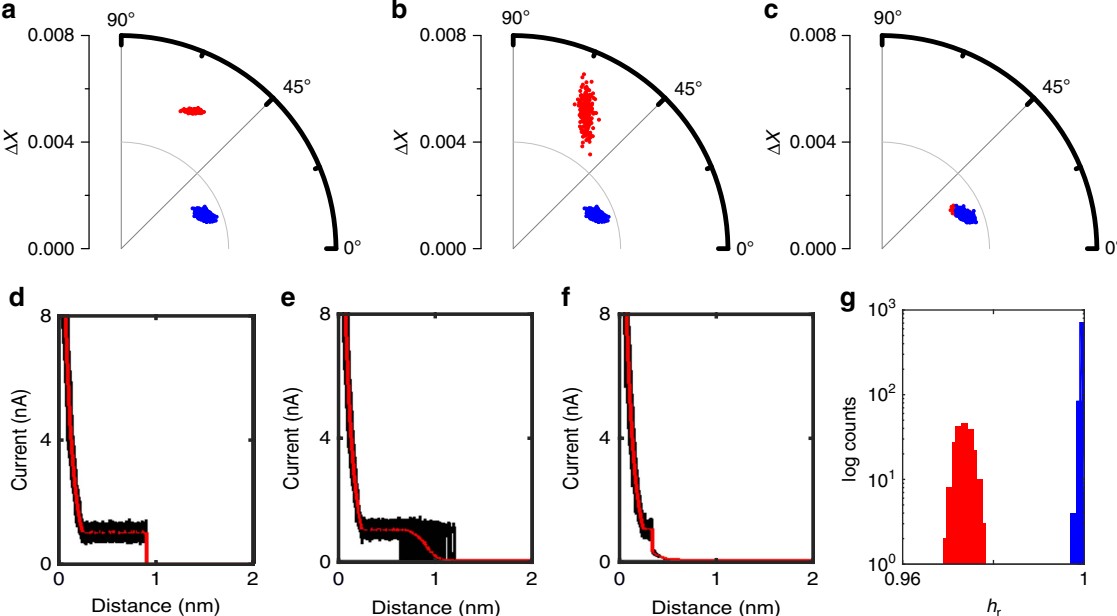

**Figure 3 | Simulated data with various $s_b$.** (**a–c**) Polar plots of data set with $s_b$ of 0.9 nm, 0.9 ± 0.1 nm STDEV and 0.35 nm. (**d–f**) corresponding overlaid plots of plateau-containing $I(s)$ traces (black) with their mean (red). (**g**) $h_r$ histogram for $s_b = 0.35$ nm data (bin width: 0.0014).

However, at high noise ($100\% \cdot I_p$), the two event clusters completely overlap in the ($\Delta X$, $\theta$) plane of the cylindrical plot (Fig. 2c) and the peak feature in the histogram (Fig. 2f) is no longer visible. However, two distinct peaks emerge between 0.8 and 1 in the $h_r$-histogram, as a third level of differentiation, and separation is still possible via MPVC, Fig. 2g (misclassification rate: 0%). The effect of multiple molecules bridging the gap at different noise levels is investigated in Supplementary Fig. 2. Consideration on the number of clusters is included in the Supplementary Figs 3–5. 2D current histograms and sample $I(s)$ traces for each noise level are included in Supplementary Figs 6 and 7.

Next, we explored the effect of changes in the break-off distance $s_b$ as well as the effect of its variance, Fig. 3. As described above, those two factors are of special interest in $I(s)$ experiments, because there is generally more than one possible way for a molecule to couple to the surface of electrodes[13]. As a result, different $s_b$ values may be observed for a given molecular system, potentially also affecting the junction conductance (*vide infra*). As mentioned above, three simulated data sets were generated, *cf*. Methods section. Each data set consisted of 1000 $I(s)$ traces, 20% of which contained plateaus with break-off distances of $s_b = 0.9$ nm, 0.9 ± 0.1 nm and 0.35 nm, respectively (noise STDEV: 0.1 nA, RoI: 0.1 to 2 nm; red data points in Fig. 3). The remaining 80% were plain exponential decays (blue). 1D, 2D histograms and sample traces are included in Supplementary Figs 8 and 9.

For $I(s)$ traces with long plateaus ($s_b = 0.9$ nm), Fig. 3a,b, the two event classes appear as distinct clusters in the polar plot and separation is trivial (either based on $\Delta X$ or $\theta$). Under these conditions, a 1D current histogram also provides the plateau current $I_p$ and hence the molecular conductance, *cf*. Supplementary Fig. 9. Once we allow for some variation around the mean break-off distance (0.9 ± 0.1 nm), Fig. 3b,e, the plateau-containing cluster spreads out, in terms of both $\Delta X$ and $\theta$, but is still well-separated from the exponential traces in the polar plot. On the other hand, in the 1D current histogram for shorter plateau lengths differentiation becomes increasingly difficult, and

essentially impossible for very short plateaus ($s_b = 0.35$ nm), *cf*. Fig. 3c,f and Supplementary Fig. 9.

In the polar plot, the individual event clusters strongly overlap in the $\Delta X$, $\theta$ dimensions, but can be clearly differentiated via $h_r$, as illustrated in Fig. 3g).

Subsequently, we investigated the ability of the vector-based method to differentiate sub-populations in the data. For this purpose, we generated 1000 $I(s)$ traces of simulated data consisting of 40% exponential decays and 20% each of simple plateaus, and plateaus with superimposed sinusoidal and telegraphic noise features (additional current noise: 0.2 nA in all cases), Fig. 4a. While the exponential decays and plain plateaus stand for through-space tunnelling and conventional molecular bridging events (see above), the sine-shaped event is a proxy for non-linear event shapes, as reported previously[12,23,22]. Telegraphic noise features have also been observed previously in $I(s)$ experiments and are typically thought to be associated with dynamic effects in the junction[11,18,21]. A polar plot showing the point density as well as the 1D current histogram are included in Supplementary Figs 10 and 11.

Such scenarios would normally be analysed using 2D current–distance histograms, since different event shapes become apparent, as long as the S/N ratio is sufficiently high. As shown in Fig. 4b, this is indeed the case, at least to some extent (note the logarithmic scale on the ordinate). However, without separating the individual traces, it would be difficult to extract the actual event characteristics or assess the relative abundance.

MPVC, on the other hand, does provide further insight into the individual event characteristics even at high low S/N levels, as shown in **c** (RoI 0.4–1 nm), and four event clusters become apparent (colour coding as in Fig. 4a). As mentioned above, we discuss the separation process in a sequential manner for illustration purposes.

First, the $\Delta X$ histogram allows for differentiation between the exponential event cluster (blue, Fig. 4d), the telegraphic plateau cluster (magenta, Fig. 4e) and a third cluster, which encompasses the sine-shaped and plain plateau features (black, Fig. 4e). The latter is then differentiated in the $\theta$ histogram, as shown in panel

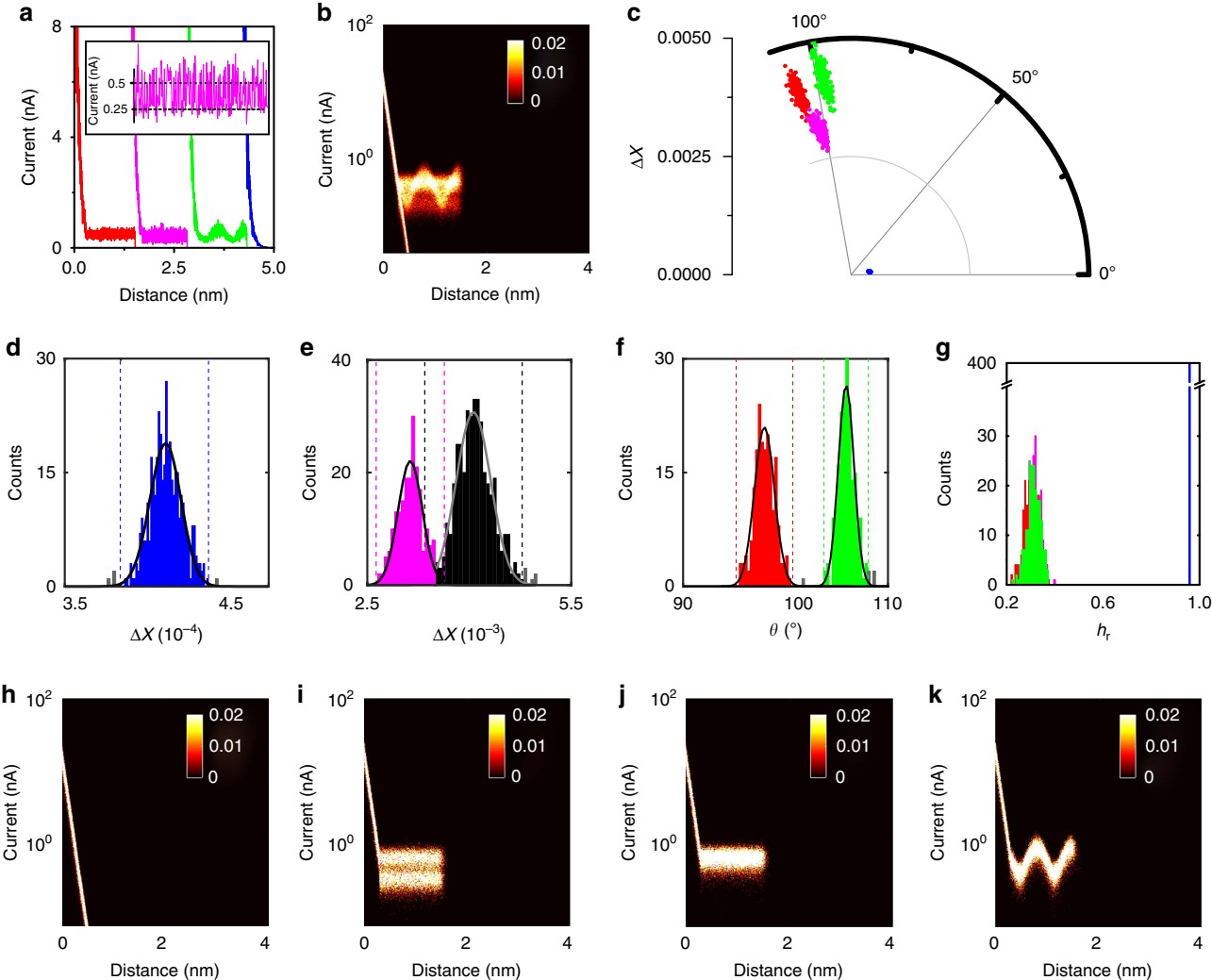

**Figure 4 | Vector-based analysis on different plateau shapes.** (**a**) Simulated $I(s)$ traces with plain (red), telegraphic (magenta), sine-shaped (green) event features, as well as exponentially decaying traces (blue). Inset: enlarged section of telegraphic switching with dotted lines indicating the 'on' and 'off' states. (**b**) All-data 2D current histogram (Current binning: 150 bins/decade, distance bin width: 0.004 nm for all 2D histograms). (**c**) Polar plot of whole data set. (**d**) $\Delta X$-histogram (bin width: $9 \times 10^{-7}$). (**e**) $\Delta X$-histogram (bin width: $5 \times 10^{-5}$) Dashed lines represent $2\sigma$ cutoffs, grey bars indicate counts outside the cutoffs. (**f**) $\theta$ histogram (Bin width: $0.3°$) (**g**) $h_r$ histogram (Bin width: $8.5 \times 10^{-3}$) (**h–k**) 2D current histograms for each event cluster after separation.

Fig. 4f. Further analysis of the data set via $h_r$ does not provide evidence for any further sub-populations, panel Fig. 4g, as expected for this simulated data set. Finally, we plot the 2D current histograms of each cluster, panels Fig. 4h–k. Each cluster is well separated and the individual event shapes can be identified. In addition, this analysis allows for the relative abundance of each event class to be determined, which would not be possible without some form of the data classification. In total, 3 $I(s)$ traces were mislabelled (0.3%), 20 $I(s)$ traces were outside the $2\sigma$ cutoffs used to define the individual event classes (2%).

**Experimental data.** After testing the vector-based approach on simulated data, we apply the methodology to the experimental data, namely from experiments with Au/ODT/Au and Au/OPE/Au. To this end, an extensive body of literature in both cases allows for an independent assessment of the results obtained here.

For ODT, 70,000 $I(s)$ traces were recorded at $V_{bias} = 0.3$ V and $I_0 = 20$ nA as described in the Methods section. (Note that this data set was previously published and analysed using a plateau identification algorithm which could however not discriminate

between different event classes[12].) The RoI was from 0.25 to 2 nm (length of ODT, in an all-extended configuration: 1.52 nm, $s_b = 0.93$ nm, accounting for tip-surface distance $s_0$ at the set point current[12]).

$\Delta X$, $\theta$ and $h_r$ of the whole data set are visualized in cylinder coordinates in Fig. 5a. The all-data 1D and 2D log current histograms in Fig. 5b,c suggest the presence of plateau features, even though the S/N ratio in both representations is relatively low. Specifically, the 1D current histogram has a shoulder, that is a peak that is partially hidden in the exponential background, which could be analysed further using appropriate data fitting. The 2D log current–distance histogram reveals a faint plateau feature at $I_p \approx 1$ nA and $s_b \approx 1$ nm.

To explore the data further, the data set was clustered into two sub-clusters using a Gustafson–Kessel Fuzzy Clustering algorithm, a generalization of the fuzzy $c$-means algorithm (FCM)[39], including covariance matrices that allows for the partitioning of ellipsoidal clusters (see Methods, Supplementary Note 1 and Supplementary Fig. 12 for limitations of cluster analysis)[40]. The 'molecular' data are contained in the red cluster (29,334 total). The blue cluster features the plain exponential traces without

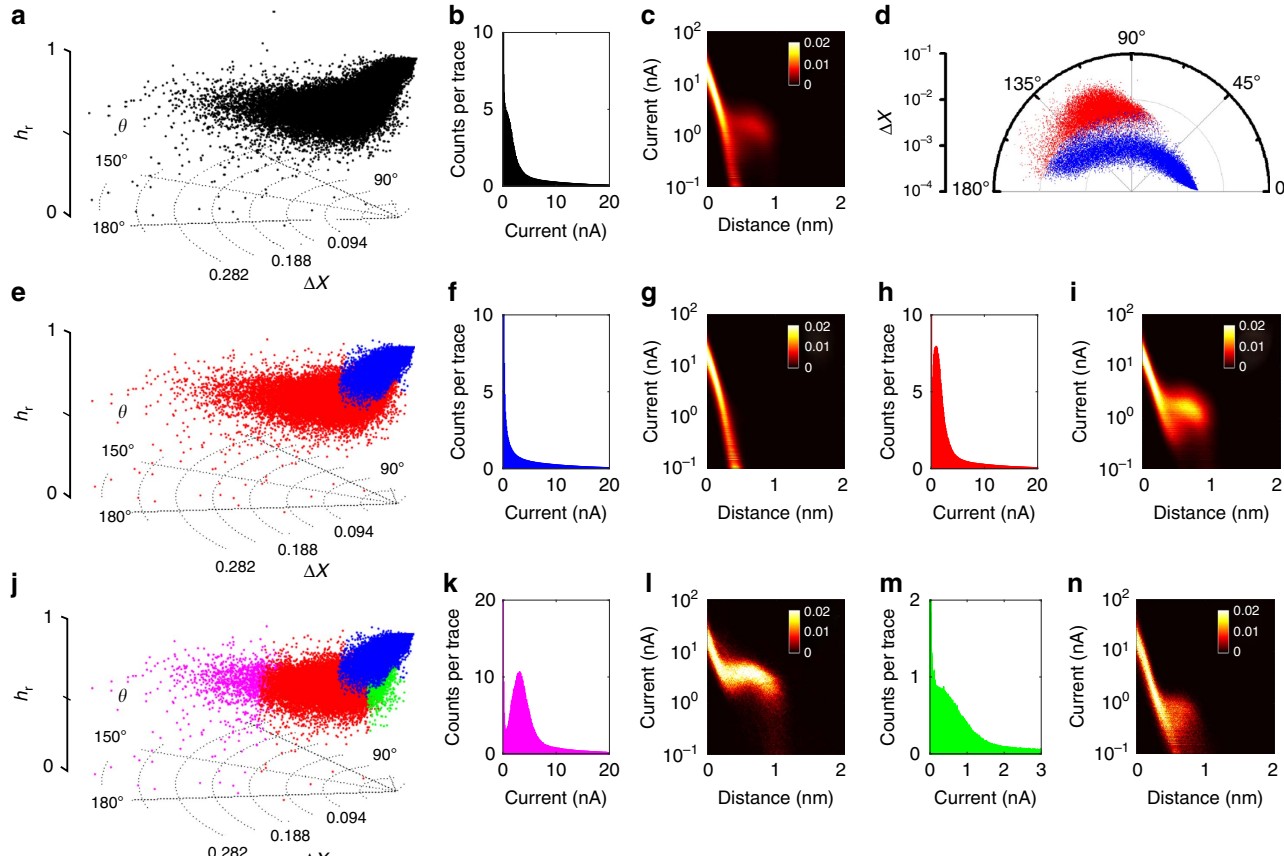

**Figure 5 | Experimental data set of ODT.** (**a**) Cylinder plot of all data points, (**e**) after Fuzzy Clustering and (**j**) with predominantly low (green, 5% lowest $\Delta X$)/ high (magenta, 5% highest $\Delta X$) conductance area indicated. (**b,c**) All the data 1D current and log 2D current–distance histogram. (**d**) Log $\Delta X$ cylinder plot, viewed along $z$ axis, to illustrate the two clusters in the data set. (**f,g**) 1D current and log 2D current–distance histograms of exponential cluster. (**h,i**) 1D current and log 2D current–distance histograms of molecular cluster. (**k,l**) 1D current and log 2D current–distance histograms of high conductance cluster. (**m,n**) 1D current and log 2D current–distance histograms of low conductance region. (Bin widths: 0.1 nA for (**b,f,h,k**) and 0.015 nA for (**m**), Current binning: 150 bins/decade, distance bin width: 0.004 nm for all 2D histograms).

plateaus (40,666 traces). A 2D log polar plot in Fig. 5c clearly shows the two distinct clusters emerging with the MPVC. In Fig. 5e, all three parameters are presented in a cylinder plot. Figure 5f,g shows the 1D and log 2D current histogram of the blue cluster, as expected, lacking any peaks or plateaus. The current histogram of the red cluster on the other hand features a clear peak at 1.0 nA corresponding to $G = 3.3$ nS ($4.3 \cdot 10^{-5}\,G_0$). This is close to the value previously reported for medium-conductance group in this system, with a most probable conductance of 3.82 nS ($4.9 \cdot 10^{-5}\,G_0$)[13].

The identification of this conductance group raised the question, whether any other conductance groups, sometimes observed for ODT and other alkane dithiols depending on experimental conditions[24,41–43], are also present in the data. For those groups, the most probable single-molecule conductance was reported to be $G = 0.9$ and 17 nS, respectively[13]. With the present choice of **R**, one would expect to find high-conductance traces towards higher $\Delta X$ as they would be further from **R**, compared with the medium-conductance group. The low-conductance group should be closer to the reference vector and therefore should be found at low $\Delta X$. As we show below, signatures of low- and high-conductance junctions are indeed found, even though they do not form distinct event clusters in the MPVC representation here (potentially due to low abundance).

To explore the low-conductance regime first, we selected the 5% of traces (1,467 total) within the red cluster that have the

lowest $\Delta X$, as shown in green in Fig. 5j. (Note: in doing so this aspect can no longer be considered 'unsupervised', but this 'sacrifice' is accompanied by valuable additional insight into the data). Compiling these data into a 1D current histogram, Fig. 5m, yields a shoulder at $I_p = 0.4$ nA (1.3 nS or $1.7 \cdot 10^{-5}\,G_0$), which is indeed similar to the previously reported value for this group. However, the relatively low $I_p$ value, its variance and the relatively low abundance of this group render its direct identification rather difficult. Figure 5n shows the corresponding 2D log current histogram.

A similar picture emerges for the junctions with high conductance. Again, we selected the 5% of molecular $I(s)$ traces with the highest $\Delta X$, Fig. 5j (magenta dots). The corresponding 1D current histogram, Fig. 5k, shows a peak at 3.15 nA (10.5 nS), which compares with 17 nS previously reported in the literature (2D log histogram shown in Fig. 5l). The reason for this discrepancy is currently unclear. However, closer inspection of the corresponding $I(s)$ traces reveals that in our data, this group rarely entirely dominates an $I(s)$ trace. Rather, we observed frequent switching between different conductance states within a given $I(s)$ trace, which prevents the emergence of well-defined clusters in the MPVC representation and may lead to shifting of peaks in the 1D current histograms. Several examples are given in Supplementary Fig. 16, even though the statistical properties and physical reason behind this switching clearly require further study. Representative $I(s)$ traces from predominantly

low, medium and high conductance region as well as a polar plot of the point density are provided in Supplementary Figs 13–15 (see also Supplementary Note 2).

The classification results above thus show that the algorithm is capable of identifying sub-populations in a large set of experimental $I(s)$ traces, in the case of ODT, as well as their relative abundance. We did not observe distinct low- and high-conductance clusters in MPVC representation under the experimental conditions used. This may be due to low abundance or potentially suggest that distinct clusters do not form in these instances and the bottom and top 5% as defined above rather represent extreme values of a large distribution. Generally, good correspondence of conductance values with data previously reported in the literature, albeit recorded in many different experiments, further supports the applicability of this approach.

Finally, we applied the MPVC algorithm to OPE, Fig. 6b, inset[44]. Its $\pi$-conjugated, rigid bridge motif has been studied in various forms in the context of single-molecule electronics over the last 20 years[45]. Although early reports presented conductance values over quite a broad range ($10^{-2}$ to $10^{-5}$ $G_0$)[46], in recent years several single-molecule studies of unfunctionalized OPE (for example, no solubilizing side groups) have converged on numbers between 1.2 to $2.9 \cdot 10^{-4}$ $G_0$ (refs 44,46–51). An additional through-molecule conductance feature at $< 10^{-5}$ $G_0$ is also sometimes observed (perhaps most clearly when the

molecule is functionalized with hexyloxy solubilizing groups)[49,52,53]. This has been attributed to conductance through two molecules of OPE interacting via $\pi$–$\pi$ stacking[52] between phenyl groups, a configuration first observed in monothiolated OPE-analogues by Wu et al.[44]. In support of this hypothesis, it is noted that the low-conductance feature exhibits a larger $s_b$ than the high-conductance plateau. To this end, we applied the above process to a data set consisting of 2,000 $I(s)$ traces (RoI: 0.4 to 4 nm based on a sulfur–sulfur distance for this molecule of 2.07 nm (ref. 44).

$\Delta X$, $\theta$ and $h_r$ of the whole data set are shown in the cylinder plot in Fig. 6a. The corresponding plot of the point density is provided in Supplementary Fig. 17. Multiple populations become visible in this representation. Accordingly, there is a shoulder in the all data 1D current histogram (Fig. 6b) as well as the faint plateau in the 2D log current histogram (Fig. 6c). In contrast to ODT, OPE shows a lower junction formation probability, which renders the assignment of single-molecule conductance values from all-data representations more difficult and data classification even more important.

By clustering $\Delta X$, $\theta$ and $h_r$ of the whole data set with a Gustafson–Kessel Fuzzy Clustering algorithm, the data split up into three clusters (Supplementary Note 3). The blue sub-group consists only of featureless exponential decays, as confirmed by the 1D current and 2D log current–distance histograms

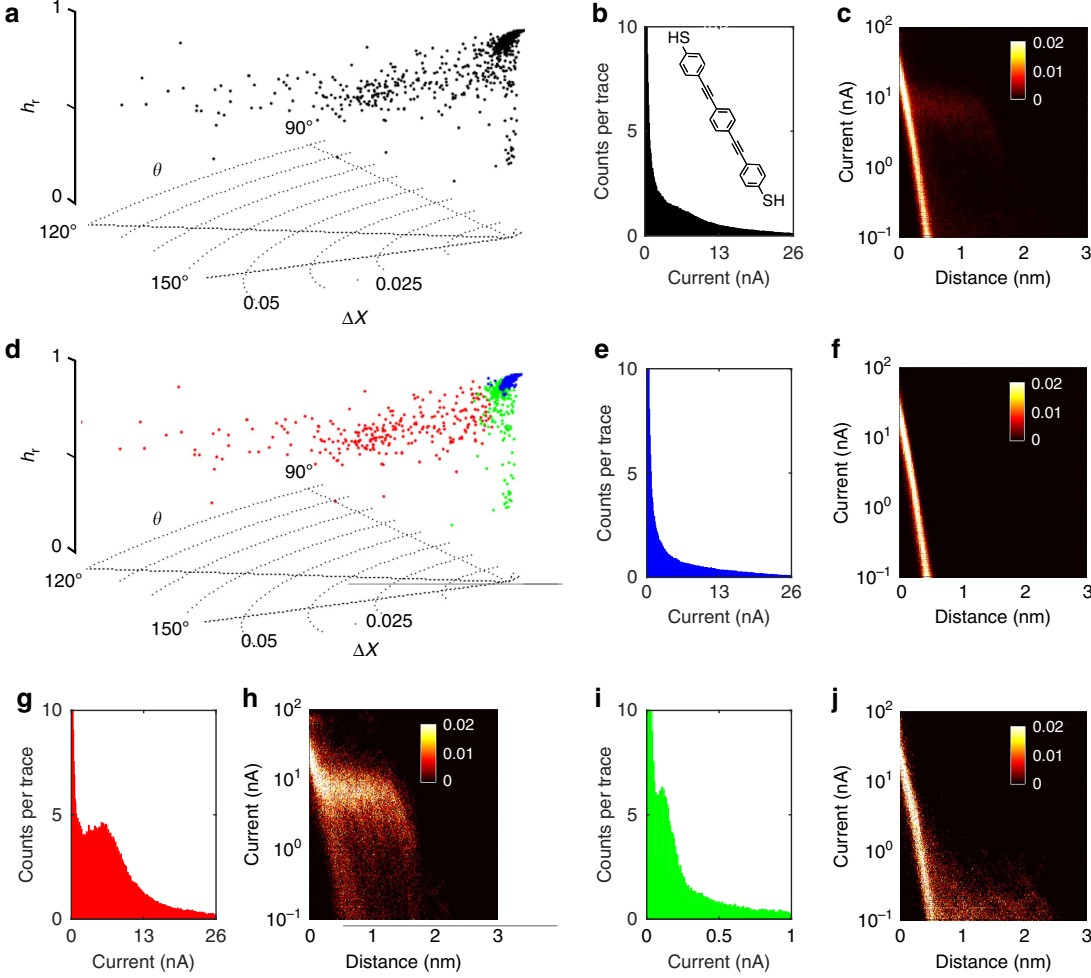

**Figure 6 | Experimental data set of OPE.** (**a**,**d**) Cylinder plots of experimental data unassigned, and after Fuzzy Clustering. (**b**) All data 1D current histogram with OPE inset (bin width 0.1 nA). (**c**) All data 2D log current histogram. (**e**,**f**): 1D current (bin width 0.1 nA) and 2D log current histogram of the plain exponential cluster. (**g**,**h**) 1D current and 2D log current histograms of the high conductance cluster (bin width 0.1 nA). (**i**,**j**) 1D current (Bin width 0.005 nA) and 2D log current histogram of the low-conductance cluster (2D log histograms: 100 bins/decade, 1,000 bins/nm).

(Fig. 6e,f), lacking any peaks or plateaus. The cluster encompasses ~74% of the total number of $I(s)$ traces.

The peak current of the red, high-conductance cluster (14% of total traces) was determined from the 1D current histogram in g) to be 5.5 nA (13.8 nS, $1.8 \cdot 10^{-4}$ $G_0$), as compared to the value of 9.3 nS ($1.2 \cdot 10^{-4}$ $G_0$) reported previously[46]. The corresponding plateau can be seen clearly in the 2D log current histogram in Fig. 6h. The colour-coded cylinder plot in Fig. 6d shows the position of each cluster.

Extracted from the green cluster in I (11% of total traces), the 1D current histogram features a low-conductance group with a current maximum at 0.11 nA, corresponding to 0.28 nS ($3.6 \cdot 10^{-6}$ $G_0$). This compares with the 'two-molecule' conductance case reported by Calame et al. with 0.46 nS (ref. 44; $5.9 \cdot 10^{-6}$ $G_0$), where the occurrence of the low conductance group is explained by aromatic ($\pi/\pi$) coupling between two molecules. Interestingly, the peak feature in i) results from a rather small number of traces (22 or 1% or the overall data set) with $h_r < 0.6$, as shown in Supplementary Fig. 21 (a small sub-cluster in the green data points can also be seen in Fig. 6d), towards low $h_r$ values). As shown in Supplementary Fig. 22, this small sub-cluster features an apparent break-off distance of 2.6 nm ($s_b$ histogram shown in Supplementary Fig. 21). Accounting for the length of the gold–thiol bonds and $s_0$, this yields a sulfur–sulfur distance of 2.7 nm, which is in very good agreement with the value of 2.91 nm, as reported by Calame et al.

The remainder of the data in the green cluster (~200 traces, 10%) appear to show a plateau-like feature, which is, however, relatively poorly defined. $s_b$ is ~0.7 nm, much shorter than the (extended) molecular length of 2.07 nm, vide supra. It is thus possible that those traces originate from junctions where the tip makes contact to the molecule in an 'off centred' configuration, that is the molecular bridge ruptures well before the tip/surface distance reaches the molecular length. Sample traces of both high- and low-conductance class are provided in Supplementary Figs 18 and 19 together with $h_r$ and 2D log current histograms of the low conductance cluster in Supplementary Figs 20–23.

As previously observed for the simulated as well as the ODT data, we were able to identify the most abundant feature in the data set with good agreement to previously reported values for $G$ and $s_b$. We also found specific, previously known sub-groups and were able to extract their relative abundance, information that would normally be inaccessible with conventional methods of analysis.

## Discussion

We have demonstrated here that multivariate pattern analysis, in particular MPVC, is a powerful tool for analysing single-conductance data. In contrast to conventional current- or current–distance histogram-based analysis, MPVC does not make any prior assumptions with regards to the shape of a molecular signature in the $I(s)$ trace, or their relative statistical abundance in the data set. Rather, the algorithm looks for similarity between data sets, in terms of vector distance, angle and (reduced) Hamming distance, relative to a common reference. The latter allows one to focus on particular event characteristics, if required, and reduces the complexity of the data significantly.

In the simulated data that closely resemble experimental $I(s)$ traces, we illustrate the general effect of different molecular characteristics on the classification parameters, such as the statistical variation in the plateau current $I_p$ or the break-off distance $s_b$. Some of these result in distinct cluster shapes, which may facilitate an initial assessment of the junction characteristics. More detailed analysis allows for extraction of different event classes as well as their relative abundance, which are inaccessible

with conventional methods of analysis, in particular for very large data sets.

Furthermore, we employed MPVC to two experimental systems with up to 70,000 $I(s)$ traces, namely Au/Au junctions of ODT and OPE, as well-characterized model systems. In both cases, we confirmed the single-molecule conductance values and break-off distances reported in the literature, notably in one data set and without hand selection of the data. We also found sub-populations in the data that would be invisible to all-data analyses, and were able to propose a link to their physical origin (for example, $\pi/\pi$ stacking interaction between OPE dimers).

More generally, the present study highlights the potential of machine learning algorithms in single-molecule science, which may provide new insight into physical processes at the nanoscale, in device or sensing applications.

## Methods

**Simulated Data.** The simulated data was generated in MATLAB (2015a) to correspond to experimental $I(s)$ data recorded at a bias voltage of 0.3 V and at a $I_0$ of 20 nA[12]. Data sets contained 1,000 individual $I(s)$ traces. Two thousand data points were produced per individual model trace at a distance between 0 and 4 nm. From experimental data, the most probable exponential decay coefficient was determined to be 1.2 Å$^{-1}$, which was used for the simulated data[12].

The simulated data was created with a plateau current $I_p$ of 1 nA (Fig. 2, Fig. 3), 0.5 nA (Fig. 4) and break-off points $s_b$ at 0.9 nm (Fig. 2) and 1.5 nm (Fig. 4). STDEVs for $I_p$ and $s_b$ were added by generating a random number with a certain STDEV around 0 and adding it to the initial value. Noise was introduced by pairwise multiplying a random vector which was distributed around 1 with a given STDEV, with the initial noise-free model vector. Data for Fig. 3 was generated with a noise STDEV of 0.1 nA. Data for Fig. 4 was generated with 0.2 nA noise STDEV, $I_p$ STDEV 0.025 and $s_b$ STDEV 0.05 nm. Telegraphic plateaus in Fig. 4 were generated by randomly switching points in the plateau between $I_p$ and $0.5 \cdot I_p$. The sine-shaped plateau $I(s)$ traces were generated by adding a sine wave with amplitude 0.2 nA and a cycle period of 0.66 nm to the plateau.

**Experimental Data.** The $I(s)$ data sets for ODT used in the present analysis were previously published by Inkpen et al., with new data for OPE collected using an identical experimental method[12]. OPE was prepared by Sonogashira cross-coupling of 4-iodophenylthioacetate and 1,4-diethynylbenzene[54]. Sub-monolayers of OPE on Au were formed by immersion of the single-crystal substrate into a 0.01 mM solution of the freshly deacetylated compound in THF for 45 s. Deacetylation was facilitated by the addition of 1 μl NH$_4$OH per 1 ml analyte solution followed by incubation for 10–15 min at room temperature[45,54]. Compared with previously reported experiments using ODT, OPE provided lower and more variable JFPs (typically ~4% based on data selection using an objective algorithm[12] where BW = 0.1, 0.2, 0.3, 0.4; PDBC > 50). Two thousand $I(s)$ traces were recorded at 26 nA $I_0$ and 0.4 V.

**MPVC.** Simulated and experimental data sets were analysed with the vector-based classification approach as described in the main text. For the analysis a noise-free reference vector was used ($I_0 = 20$ nA for ODT and 26 nA for OPE, decay coefficient $\beta = 1.0$ Å$^{-1}$), cf. Supplementary Note 4. The data set and reference were divided by the scanner range (100 nA) multiplied by the square root of the number of data points in the region of interest to normalize, cf. equation (1). This ensured that the maximal possible separation between points was unity. $s_b$ was calculated by taking 3 σ of the distribution of the last 1 nm of the individual $I(s)$ traces as a threshold. $s_b$ was considered the distance of the first current value below this threshold starting from the beginning of the trace. Fuzzy Clustering was performed with the 'Clustering and Data Analysis Toolbox' by J. Abonyi et al. downloaded from MATLAB file exchange.

**Computer code availability.** All relevant scripts are available from the authors on request.

**Data availability.** All relevant data are available from the authors on request.

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

## Acknowledgements

The authors would like to thank The Leverhulme Trust (M.L., M.S.I., N.J.L., T.A.) and the Wellcome Trust (K.K.) for funding.

## Author contributions

All the authors contributed to the writing of the article and interpretation of the results.

## Additional information

**Competing financial interests:** The authors declare no competing financial interests.

