## [Peer Review File · Nature Communications]

Reviewer #1 (Remarks to the Author):

This manuscript reported a method to analyze single-molecule charge transport data using a multi-parameter vector based classification process (MPVC). The authors applied their method to simulated and experimental data of ODT and OPE molecular systems. I feel the manuscript rather difficult to follow, particularly I have following questions:

1) The method used three parameters: length of difference vector from reference vector (ΔX), the angle (θ) between X_m and $-R$, the reduced Hamming distance (h_r). Here, the definition of the reduced Hamming distance (h_r) is not clear. The sentence of "the minimum relative number of component exchanges that render the vectors identical." should be described more concretely. What are the physical meanings of θ and h_r in this MPVC method. Authors may need to explain further in terms of how these parameters can classify each cluster.

2) I don't understand if the expression of θ in the sentence of "the angle θ between X_m and $-R$, eq. (2)" (page 4) is consistent with the expression in Figure 1(E) or not. Authors need to carefully check this.

3) I don't understand the sentence "By clustering ΔX , θ and h_r of the whole data set with a Gustafson-Kessel Fuzzy Clustering algorithm, the data split up into three clusters." (page 16). Further and more clear explanation may be needed.

4) It is not clear to me that the main difference of this manuscript from the authors' previous paper (Ref. 12. 12 M. S. Inkpen, M. Lemmer, N. Fitzpatrick, D. Costa-Milan, R. J. Nichols, N. J. Long and T. Albrecht, J. Am. Chem. Soc., 2015, 9971-9981).

In summary, Although this manuscript was hard to follow and understand, this manuscript would be helpful in interpreting and analyzing STM-based single molecule transport data. In this sense, I don't have objection against this manuscript. But I hope that the authors make more efforts to explain better for general readers.

Reviewer #2 (Remarks to the Author):

This manuscript describes a data processing/analysis algorithm (MPVC) based multi-parameter vector-based classification for STM-break junction based single molecule conductance measurement. The algorithm, if proven generally applicable to different systems, would have a value to the community of molecular electronics. I believe the work may be publishable after addressing the following points.

(1) MPVC classifies data sets by looking for similarities. However it is unclear that similar data sets represent true experimental observations with meaningful physics. In other words, it is unclear how to assign each separated event cluster to a meaningful physical process without prior knowledge about the system.

(2) Only two examples are given, and more examples will be needed to show that the algorithm has a value to the molecular electronics community.

(3) To convince that the algorithm has a unique value (not just another way to presenting and analyzing data), the authors should demonstrate cases where the common approaches fail (e.g., 1D, 2D conductance vs. distance histograms in linear and logarithmic scales), and but MPVC works.

Other minor suggestions:

(1) There is a need to describe Eq. 2 to help readers to understand it.

(2) I have trouble understand Fig. 4DI. The description on page 11 and figure caption are not clear. Also the figure caption of Fig. 5 is also confusing.

Reviewer #3 (Remarks to the Author):

This manuscript demonstrated the multivariate pattern analysis for simulated and experimental single-molecule charge transport data. The results are solid and the manuscript is well written. I would like to recommend to acceptance for the publication. To improve the manuscript more, I have following minor technical comments.

1. The multi-variate pattern analysis is an interesting tool, because it does not make any prior assumptions with regards to the shape of a molecular signature in the I(s) trace. I pretty much appreciate this aspect of the authors' work. However, I would like to ask: Do the specific experimental achievements and findings reported here really justify publication in a journal geared towards a very broad audience such as Nature Communications?

2. In this paper, the author have used the thiol-terminated molecule (1,8-octanedithiol; ODT) for the single-molecule charge transport measurements. However, according to the references such as J. Am. Chem. Soc. (2012), 134, 4003, they demonstrated that the strong Au–thiol bond can cause structural rearrangements in the electrodes with varied atomic structure that sustain a broad range of conductance values, which do not show a well-defined molecular conductance signature. What is the author's opinion about the unstable Au-thiol bond effect on the multi-variate pattern analysis? I wonder if author can use multi-variate pattern analysis for the amine or methylsulfidated (which bind to Au through a donor-accept bond more stably) terminated molecules formed with Au electrodes rather the thiol group.

3. The experimental data set in Figure 5. shows the high and low conductance region for the ODT-Au molecular junctions. I don't quite see why one would expect the two different conductance groups for the same molecular junctions. This is an interesting result. However, there is no theoretical backup offered in support of these results. In particular, what are the geometrical structures for the high and low conductance junctions formed with Au atoms and what makes the conductance as well as the molecular plateau length be different even the same ODT-Au junctions?

"Unsupervised Vector-based Classification of Single-Molecule Charge Transport Data" (revision 01)

Point-by-point response to the referees

Reviewer #1 (Remarks to the Author):

This manuscript reported a method to analyze single-molecule charge transport data using a multi-parameter vector based classification process (MPVC). The authors applied their method to simulated and experimental data of ODT and OPE molecular systems. I feel the manuscript rather difficult to follow, particularly I have following questions:

1) The method used three parameters: length of difference vector from reference vector (Del_X), the angle ($theta_m$) between X_m and $-R$, the reduced Hamming distance (h_r). Here, the definition of the reduced Hamming distance (h_r) is not clear. The sentence of "the minimum relative number of component exchanges that render the vectors identical." should be described more concretely. What are the physical meanings of $theta_m$ and h_r in this MPVC method. Authors may need to explain further in terms of how these parameters can classify each cluster.

We have taken on board the impression of this (and other) referee(s), who found the text difficult to follow in places. We feel this may have to do with the conceptually rather different approach that MPVC takes to look $I(s)$ and potentially other data. In response, we have made alterations throughout the text, in the hope of making the content more accessible (see 'changes in red' version of the revised manuscript).

We have modified figure 1 and replaced two panels (D and E). In D), we now show a flow chart intended to illustrate the 'philosophy' behind vector-based classification, in comparison with conventional approaches (as discussed in the main text). E) has now been updated to better reflect the graphical properties of the vectors (the equation and the figure also seem to be in agreement).

Moreover on pages 4-6, we have completely re-written the way we describe the classification parameters, how those relate to the actual $I(s)$ data, and added a discussion around the capabilities and limitations of vector-based classification. This section now reads (pp. 4):

"We then calculate three vector properties in relation to the data sets X_m and R , namely, the length of the difference vector $|Y_m| = \Delta X = |X_m - R|$; the angle θ_m between Y_m and $-R$, eq. (2),

$$\cos(\Theta_m) = -\frac{R \cdot Y_m}{|R| \cdot |Y_m|} \quad (2)$$

and the reduced Hamming distance h_r between vectors $Y_{r,m} = Y_m/\text{abs}(Y_m)$ and $R/\text{abs}(R)$. The Hamming distance is the minimum relative number of component exchanges that render two vectors identical.³⁴ That is, if two vectors are identical, then the Hamming distance is 0 (no change required); if all components differ, then the Hamming distance is 1 (every component in one vector needs to be changed). Note that $\text{abs}(Y_m)$ and $\text{abs}(R)$ are vectors containing the absolute component values of Y_m and R , i.e. they are not the length of the vectors Y_m and R , respectively. The component values of $Y_{r,m}$ and $R/\text{abs}(R)$ are thus either -1, 0 or 1."

Note that eq. (2) is standard in vector algebra to determine the angle between two arbitrary vectors, so we have referenced this explicitly."

We then expand on their relation to the actual $I(s)$ data:

"In relation to the $I(s)$ data, these three parameters ΔX_m , θ_m and $h_{r,m}$ may be illustrated in the following way: ΔX_m is a measure for the total variation of a given $I(s)$ trace relative to the reference trace R , as described above. No distinction is made between positive and negative deviations from R and thus two differently shaped $I(s)$ traces could feature the same ΔX_m value. For example, current values in a given trace 1 may be slightly higher than R over all s , while values in a second trace 2 may be significantly higher than R for low s and then much lower than R at large s . However, the dot product between R and Y helps to differentiate these two cases, since positive and negative vector components cancel out. Thus, $R \cdot Y_1 \neq R \cdot Y_2$, resulting in different angles θ_1 and θ_2 between Y_1 and Y_2 , and R in vector space (the denominator in eq. (2) serves to normalize with regards to the vector lengths). Finally, h_r helps to quantify, how large a fraction of the $I(s)$ values in a given trace lies above or below R . For example, a third $I(s)$ trace may oscillate around R in a way that $\Delta X_2 = \Delta X_3$ and $\theta_2 = \theta_3$, but with more data points above R than trace 2 (e.g. $Y_{r,3}$ has more '1's than $Y_{r,2}$). Thus, $h_{r,2} \neq h_{r,3}$ and the two curves can be differentiated."

We also put the choice of parameter/classifiers into context and discuss the strength of MPVC - as well as what it cannot deliver:

"It should be stressed that the above choice of parameters, while sensible, is neither complete or exhaustive. Other statistical properties of the $I(s)$ traces, such as the centre-of-mass of the curve, could be used to in addition or instead of one of the three parameter defined above. Generally, if the number of classifiers is too small, the classification will lack specificity and a cluster in the multi-parameter representation will contain many different types of $I(s)$ traces. On the other hand, a too large number of classifiers may result in a large dispersion in the multi-parameter representation, rendering the identification of clusters more difficult. We found that the combination of ΔX_m , θ_m and $h_{r,m}$ struck a good balance in this regard.

An important feature of the above MPVC algorithm, compared to conventional methods of analysis, is that it does not make any *a priori* assumptions with regards to the signal shape, e.g. whether a plateau feature is present or not. Rather, it looks for similarities between the measured data sets, relative to R and based on the classifiers used. A large number of similar traces will thus produce regions of high point density in the $(\Delta X, \theta, h_r)$ representation, which can then be clustered and processed for further analysis.

Notably, MPVC does not provide the physical interpretation. It is partly for this reason that we chose both complex simulated data and well-characterized experimental system, where the interpretation is (largely) known, as a benchmark for MPVC here. The former allows for detailed characterization of the classification results (misclassification, especially where there are multiple groups), while the latter facilitates the physical interpretation of the clusters.

Finally, we note that by using a common reference vector we avoid having to compute the pair-wise distance matrix for all $I(s)$ traces to determine similarity between traces. This effectively reduces the dimensionality of the data and is a significant advantage for large data sets (as for the **ODT** data with 70,000 traces, see below)."

2) I don't understand if the expression of θ_m in the sentence of "the angle θ_m between X_m and $-R$, eq. (2)" (page 4) is consistent with the expression in Figure 1(E) or not. Authors need to carefully check this.

We thank the referee for checking the manuscript so carefully. Indeed, we found a misprint in the formula, which has now been corrected in the revised version of the manuscript.

3) I don't understand the sentence "By clustering ΔX , θ and h_r of the whole data set with a Gustafson-Kessel Fuzzy Clustering algorithm, the data split up into three clusters." (page 16). Further and more clear explanation may be needed.

Gustafson-Kessel Fuzzy Clustering is a well-documented clustering method in the literature and we have provided a suitable key reference. In response to the referee comment, we have expanded on GK Fuzzy Clustering in the main text (page 13, in the section on ODT, not in the later part on OPE), as well as (more substantially) in the SI (pages 5-6). We have also some justification in the text as to why and how we used this clustering method. k-means and k-medoid clustering did not give satisfactory results, due to the sometimes highly asymmetric cluster shapes.

We recognize that generally a complicating factor is that the number of clusters must be provided to the GK Fuzzy clustering algorithm. This is still a fundamental problem in Machine Learning, which we cannot attempt to solve here.

The cluster number can be justified in different ways, namely by visual methods (clusters can be obvious to the human eye, but we chose not to use this approach), by distribution-based method (an approach we have taken in a number of cases) or by semi-empirical methods (“elbow method”, where the cost function is plotted versus the number of clusters). Alternatively, density based clustering methods like the one described by Rodriguez and Liao in Science in 2014 (our ref. 35) can be used, which avoid the problem of the number of clusters. However, here the user manually has to make a decision of the density cut-off, so there is significant user intervention as well.

4) It is not clear to me that the main difference of this manuscript from the authors' previous paper (Ref. 12. 12 M. S. Inkpen, M. Lemmer, N. Fitzpatrick, D. Costa-Milan, R. J. Nichols, N. J. Long and T. Albrecht, J. Am. Chem. Soc., 2015, 9971-9981).

The paper published by Inkpen *et al.* relies on prior assumptions of the data set, namely on the expectation that an $I(s)$ trace forms a plateau, while the molecular junction is intact. In that sense, the Inkpen *et al.* method is 'conventional' in that it employs current histograms to extract molecular data. This makes sense if indeed the $I(s)$ traces feature a well-defined plateau, but breaks down if the event shape is different (What is the meaning of a maximum in the current histogram, if the 'plateau' current fluctuates?). As we point out in the main text, there well-documented cases in the literature (both theoretical and experimental studies, inc. the paper cited above) showing that event shapes can be much more complex. This is why we feel the MPVC method is a conceptual departure from the conventional methods of analysis.

In summary, Although this manuscript was hard to follow and understand, this manuscript would be helpful in interpreting and analyzing STM-based single molecule transport data. In this sense, I don't have objection against this manuscript. But I hope that the authors make more efforts to explain better for general readers.

With the above alterations in mind, we hope we have improved the manuscript sufficiently to improve readability of the paper and allay the concerns of this referee.

Reviewer #2 (Remarks to the Author):

This manuscript describes a data processing/analysis algorithm (MPVC) based multi-parameter vector-based classification for STM-break junction based single molecule conductance measurement. The algorithm, if proven generally applicable to different systems, would have a value to the community of molecular electronics. I believe the work may be publishable after addressing the following points.

(1) MPVC classifies data sets by looking for similarities. However it is unclear that similar data sets represent true experimental observations with meaningful physics. In other words, it is unclear how to assign each separated event cluster to a meaningful physical process without prior knowledge about the system.

We would not claim that nothing needs to be known about the data that is measured. Crucially, MPVC does not make any assumptions to what represents an 'event'. We define an event as a feature in the data that occurs with a significant probability - just what this event looks like is unimportant initially. Once clusters form in the MPVC representation, these are then identified by clustering methods and then interpreted using either prior or new knowledge. We use prior knowledge, since we need to test and benchmark this new approach. Interpretation of data from new molecules would normally require additional experimental characterization and calculations, but this is beyond the scope of the present paper. But this does not mean that every cluster will have physical meaning (e.g., in the case where the tunnelling current saturates quickly, due to extensive drift) - all that MPVC does is to check whether particular features repeatedly appear in the data.

We have amended the revised version of the manuscript with the following text on page 6, to clarify what we think MPVC can deliver and what it cannot (see also comments to referee 1 above):

"It should be stressed that the above choice of parameters, while sensible, is neither complete or exhaustive. Other statistical properties of the $I(s)$ traces, such as the centre-of-mass of the curve, could be used to in addition or instead of one of the three parameter defined above. Generally, if the number of classifiers is too small, the classification will lack specificity and a cluster in the multi-parameter representation will contain many different types of $I(s)$ traces. On the other hand, a too large number of classifiers may result in a large dispersion in the multi-parameter representation, rendering the identification of clusters more difficult. We found that the combination of ΔX_m , θ_m and $h_{r,m}$ struck a good balance in this regard.

An important feature of the above MPVC algorithm, compared to conventional methods of analysis, is that it does not make any *a priori* assumptions with regards to the signal shape, e.g. whether a plateau feature is present or not. Rather, it looks for similarities between the measured data sets, relative to R and based on the classifiers used. A large number of similar traces will thus produce regions of high point density in the $(\Delta X, \theta, h_r)$ representation, which can then be clustered and processed for further analysis.

Notably, MPVC does not provide the physical interpretation. It is partly for this reason that we chose both complex simulated data and well-characterized experimental system, where the interpretation is (largely) known, as a benchmark for MPVC here. The former allows for detailed characterization of the classification results (misclassification, especially where there are multiple groups), while the latter facilitates the physical interpretation of the clusters."

(2) Only two examples are given, and more examples will be needed to show that the algorithm has a value to the molecular electronics community.

(3) To convince that the algorithm has a unique value (not just another way to presenting and analyzing data), the authors should demonstrate cases where the common approaches fail (e.g., 1D, 2D conductance vs. distance histograms in linear and logarithmic scales), and but MPVC works.

We would like to address these two, closely related points together, which we hope is acceptable.

We agree to some extent, in that the true value of this new approach will have to be tried and tested by the community. In fact, we very much want the community to use MPVC or related vector-based methods. To this end, we have now added a comment in the Acknowledgements section, offering to make the MATLAB scripts available upon request.

At the same time, we feel we have already demonstrated a number of features of MPVC and have shown that it succeeds, where conventional methods of analysis fail. This includes

- On page 11, where we show that for very short plateaus (simulated data, fig. 3), exponential decays and plateau-containing curves in the 1- and 2D-histograms are increasingly difficult to discern. On the other hand, separation is straightforward with MPVC, especially after invoking the h_r classifier (panel C).
- On page 13, in the context of fig. 4, where we show that different types of events (shown in panel A) completely overlap in the 2D histogram, but are easily separated using MPVC (further examples in the SI).
- On page 16 (fig. 5, experimental ODT data), where sub-populations are invisible in the all-data point histogram, but differentiated after MPVC classification (panels E-G II).
- On page 19 (fig. 6, experimental OPE data), we successfully detect a low-conductance configuration, which compares well with recent results from Calame et al. (our ref. 46)

Thus, we very much think we have demonstrated the capabilities of MPVC in principle (using simulated data) and in actual fact (using experimental data). We are in the process of applying MPVC to new molecules, with significant success, but here the interpretation needs to be backed up by additional experimental method as well as theory and simulation. This is clearly beyond the scope of the present paper, given that we have demonstrated the validity of the approach as stated above.

Other minor suggestions:

(1) There is a need to describe Eq. 2 to help readers to understand it.

This has been addressed in the revised version of the manuscript.

(2) I have trouble understand Fig. 4DI. The description on page 11 and figure caption are not clear. Also the figure caption of Fig. 5 is also confusing.

Both captions and the text have been changed accordingly, as indicated in the 'changes in red' version of the revised manuscript. We hope the text is now clearer and more accessible.

Reviewer #3 (Remarks to the Author):

This manuscript demonstrated the multivariate pattern analysis for simulated and experimental single-molecule charge transport data. The results are solid and the manuscript is well written. I would like to recommend to acceptance for the publication. To improve the manuscript more, I have following minor technical comments.

We appreciate the supportive comments of this referee and address the specific comments below.

1. The multi-variate pattern analysis is an interesting tool, because it does not make any prior assumptions with regards to the shape of a molecular signature in the $I(s)$ trace. I pretty much appreciate this aspect of the authors' work. However, I would like to ask: Do the specific experimental achievements and findings reported here really justify publication in a journal geared towards a very broad audience such as Nature Communications?

Yes, we very much think so. MPVC marks a significant conceptual departure from previous analysis methods in the field of molecular electronics and enables a much deeper level of analysis and thus new physical insight. Moreover, vector-based approaches such as MPVC are not restricted to $I(s)$ data, even though this is the subject here. In the same way, vector-based approaches could be applied to other types of single-molecule charge transport data. For the first time, machine learning has been used to analyse single-molecular conductance data and even single-molecule data in general. This has not been explored so far, but should clearly appeal to a very wide audience. Equally, our study will be likely to attract interest from the Machine Learning community, which has so far not engaged with these areas at all.

2. In this paper, the author have used the thiol-terminated molecule (1,8-octanedithiol; ODT) for the single-molecule charge transport measurements. However, according to the references such as J. Am. Chem. Soc. (2012), 134, 4003, they demonstrated that the strong Au–thiol bond can cause structural rearrangements in the electrodes with varied atomic structure that sustain a broad range of conductance values, which do not show a well-defined molecular conductance signature. What is the author's opinion about the unstable Au-thiol bond effect on the multi-variate pattern analysis? I wonder if author can use multi-variate pattern analysis for the amine or methylsulfied (which bind to Au through a donor-accept bond more stably) terminated molecules formed with Au electrodes rather the thiol group.

The question of whether different conductance groups exist for Au/thiol/Au junctions is a complex one and it is the author's view that experimental results will most likely depend on experimental conditions (e.g. type of surface modification, potentially the density of the adsorbate, solution vs. air vs. vacuum), the experimental method used (break-junction vs. $I(s)$ vs. $I(t)$ technique and others, in particular with regards to the speed of junction formation). The work of Nichols, Tao and Wandlowski seems to point in this direction. Bond formation between the thiols and the Au surfaces most likely occurs in distinct locations, but the referee is correct in that atomic re-arrangements may blur conductance characteristics, depending on the conductance of extended gold contacts.

In any case, in our data set we do not see strong evidence for distinct cluster formation, even though there are a small number of traces that indeed show similar features. We have re-balanced the discussion in the manuscript to remain as unbiased as possible in relation to this question, in particular since our study was not originally designed to address this question.

To this end, on page 14/15, in order to avoid a priori suggestion as to whether such groups exist or not, we write in the revised manuscript: "The identification of this conductance group raised the question, whether any other conductance groups, sometimes observed for ODT and other alkanedithiols depending on experimental conditions,^{24,42,43} are also present in the data (for a counter example, see 44). For those groups, the most probable single-molecule conductance was reported to be $G = 0.9$ nS and 17 nS, respectively.¹³ With the present choice of R, one would expect to find high-conductance traces towards higher ΔX as they would be further from R, compared to the medium-conductance group. The low conductance group should be closer to the reference vector and therefore should be found at low ΔX . As we show below, signatures of low- and high-conductance junctions are indeed found, even though they do not form distinct event clusters in the MPVC representation here (potentially due to low abundance)."

On page 16, we now state: "We did not observe distinct low- and high-conductance clusters in MPVC representation under the experimental conditions used. This may be due to low abundance or potentially suggest that the distinct cluster do not form in these instances and the bottom and top 5% as defined above rather represent extreme values of a large distribution."

As for the second point, namely whether MPVC can be applied to other types of molecular junctions, the answer is 'yes' - the algorithm is 'blind' in this regard and we do not see why it should be in any way restricted to particular molecular anchor groups or bridges.

3. The experimental data set in Figure 5. shows the high and low conductance region for the ODT-Au molecular junctions. I don't quite see why one would expect the two different conductance groups for the same molecular junctions. This is an interesting result. However, there is no theoretical backup offered in support of these results. In particular, what are the geometrical structures for the high and low conductance junctions formed with Au atoms and what makes the conductance as well as the molecular plateau length be different even the same ODT-Au junctions?

Different conductance classes have been observed by a number of different groups in the past. (See J. Phys. Chem. C, Vol. 113, No. 14, 2009; J. Am. Chem. Soc., Vol. 130, No. 1, 2008 and references herein) Most commonly described are low, medium and high conductance groups and integer multiples of these, attributed to several molecules forming a junction. The low conductance group has been assigned to gauche defects in the molecule, the medium group to the all-trans conformation where at least one anchoring sulphur is bound in the atop configuration to a gold atom. The high conductance group is assigned to the case where both anchoring sulphur atoms are coordinated to two gold atoms in a bridge geometry.

As we stress in the manuscript, we do see some indications of these different conductance states, but we do not find distinct clusters in the MPVC representation (see above).

The difference in conductance is, among other things, thought to arise from different electronic coupling between the thiol groups and the gold (which in turn is dependent on the actual binding site, the tilt angle, and the type of molecule). Kuznetsov, Kornyshev and Ulstrup have also shown theoretically that thermally activated tunnelling in alkanedithiols can account for the length observed by Haiss et al. (cf. ref 42).

Reviewer #1 (Remarks to the Author):

(No comments).

Reviewer #2 (Remarks to the Author):

The authors have addressed the questions raised by this reviewer. That manuscript is ready for publication.

Reviewer #3 (Remarks to the Author):

This manuscript demonstrated the multivariate pattern analysis for simulated and experimental single-molecule charge transport data. The results are solid and the manuscript is well written. I would like to recommend to acceptance for the publication. To improve the manuscript more, I have following minor technical comments.

1. The multi-variate pattern analysis is an interesting tool, because it does not make any prior assumptions with regards to the shape of a molecular signature in the I(s) trace. I pretty much appreciate this aspect of the authors' work. However, I would like to ask: Do the specific experimental achievements and findings reported here really justify publication in a journal geared towards a very broad audience such as Nature Communications?

2. In this paper, the author have used the thiol-terminated molecule (1,8-octanedithiol; ODT) for the single-molecule charge transport measurements. However, according to the references such as J. Am. Chem. Soc. (2012), 134, 4003, they demonstrated that the strong Au–thiol bond can cause structural rearrangements in the electrodes with varied atomic structure that sustain a broad range of conductance values, which do not show a well-defined molecular conductance signature. What is the author's opinion about the unstable Au-thiol bond effect on the multi-variate pattern analysis? I wonder if author can use multi-variate pattern analysis for the amine or methylsulfied (which bind to Au through a donor-accept bond more stably) terminated molecules formed with Au electrodes rather the thiol group.

3. The experimental data set in Figure 5. shows the high and low conductance region for the ODT-Au molecular junctions. I don't quite see why one would expect the two different conductance groups for the same molecular junctions. This is an interesting result. However, there is no theoretical backup offered in support of these results. In particular, what are the geometrical structures for the high and low conductance junctions formed with Au atoms and what makes the conductance as well as the molecular plateau length be different even the same ODT-Au junctions?